# Mortality and its associated factors in transfused patients at a tertiary hospital in Uganda

Clement D. Okello [ID]¹*, Andrew W. Shih², Bridget Angucia¹, Noah Kiwanuka³, Nancy Heddle⁴, Jackson Orem¹, Harriet Mayanja-Kizza [ID]⁵

1 Uganda Cancer Institute, Kampala, Uganda, 2 Department of Pathology and Laboratory Medicine, University of British Columbia, Vancouver, BC, Canada, 3 School of Public Health, College of Health Sciences, Makerere University, Kampala, Uganda, 4 Department of Pathology and Molecular Medicine, McMaster University, Hamilton, Canada, 5 Department of Medicine, College of Health Sciences, Makerere University, Kampala, Uganda

* clement.okello@uci.or.ug

**Data Availability Statement:** All relevant data are within the manuscript and its Supporting Information files.

## Abstract

Blood transfusion is life-saving but sometimes also associated with morbidity and mortality. There is limited data on mortality in patients transfused with whole blood in sub-Saharan Africa. We described the 30-day all-cause mortality and its associated factors in patients transfused with whole blood to inform appropriate clinical intervention and research priorities to mitigate potential risks. A retrospective study was performed on purposively sampled patients transfused with whole blood at the Uganda Cancer Institute (UCI) and Mulago hospital in the year 2018. Two thousand twelve patients with a median (IQR) age of 39 (28–54) years were enrolled over a four month period. There were 1,107 (55%) females. Isolated HIV related anaemia (228, 11.3%), gynaecological cancers (208, 10.3%), unexplained anaemia (186, 9.2%), gastrointestinal cancers (148, 7.4%), and kidney disease (141, 7.0%) were the commonest diagnoses. Most patients were transfused with only one unit of blood (n = 1232, 61.2%). The 30 day all-cause mortality rate was 25.2%. Factors associated with mortality were isolated HIV related anaemia (HR 3.2, 95% CI, 2.3–4.4), liver disease (HR 3.0, 95% CI, 2.0–4.5), kidney disease (HR 2.2, 95% CI, 1.5–3.3; p<0.01), cardiovascular disease (HR 2.9, 95% CI, 1.6–5.4; p<0.01), respiratory disease (HR 3.0, 95% CI 1.8–4.9; p<0.01), diabetes mellitus (HR 4.1, 95% CI, 2.3–7.4; p<0.01) and sepsis (HR 6.2, 95% CI 3.7–10.4; p<0.01). Transfusion with additional blood was associated with survival (HR 0.8, 95% CI 0.7–0.9, p<0.01). In conclusion, the 30-day all-cause mortality was higher than in the general inpatients. Factors associated with mortality were isolated HIV related anaemia, kidney disease, liver disease, respiratory disease, cardiovascular disease, diabetes mellitus and sepsis. Transfusion with additional blood was associated with survival. These findings require further prospective evaluation.

**Funding:** This work was funded by the Uganda Cancer Institute - ADB research project. The funders had no role in study design, data collection and analysis, decision to publish, or preparation of the manuscript.

**Competing interests:** The authors have declared that no competing interests exist.

## Background

Blood transfusion can be life-saving in many situations. However, transmission of infections like hepatitis B and C, human immunodeficiency virus (HIV) [1], bacteria [2] and malaria [3], as well as immunomodulation mar this crucial intervention. The underlying disease and severity of anaemia in patients requiring transfusion may often be associated with increased morbidity and mortality [4, 5]. Anaemia and impaired immunity is associated with decreased survival and progression of the underlying disease [6]. Blood transfusion has also been associated with increased mortality [7]. Most adverse effects of blood transfusion are attributed to the donor leucocytes [8], abundant especially in non-leucoreduced whole blood. In high income countries, whole blood transfusion is reserved for patients with haemorrhagic shock [9] while transfusions in sub-Saharan Africa (SSA) are mainly done using whole blood [10]. A recent trial in SSA reported lower haemoglobin recovery in children transfused with packed cells than those transfused with whole blood [11].

Blood transfusion in SSA is mainly associated with cancer (33.5%), pregnancy(12.4%) and sickle cell disease (6.9%) [12]. However, limited blood supply in SSA is a major challenge [13–15] to the extent that most hospital blood banks issue blood to the patients as soon as they receive the stock. Therefore, most blood transfusion in these settings are with blood that is less than 14 days old [16]. A randomized, controlled trial conducted in six hospitals in North America, Australia and Israel [17] and meta-analyses have shown a non-significant trend towards increased mortality in patients transfused with red blood cells (RBC) stored for a shorter time compared to those stored for a longer time [18, 19]. Additionally, differences in testing strategies for pathogens and processing of blood in SSA may explain a higher incidence of acute transfusion reactions and the bacterial contamination observed in the SSA than in high income countries [2, 20]. These challenges may contribute to morbidity and mortality in transfused patients in SSA.

There is, however scant data on mortality outcomes of transfusion in most SSA countries. We, therefore, undertook a study to describe the mortality and its associated factors in patients transfused with whole blood in a tertiary hospital in Uganda.

## Methods

### Study design and setting

This was a retrospective study conducted at the Uganda Cancer Institute (UCI) and the internal medicine unit of Mulago National Referral Hospital (MNRH). Patients from the surgical wards and obstetrics/gynaecology wards were not included. Uganda Cancer Institute is the tertiary cancer treatment facility, while MNRH is the national referral and teaching hospital, and are all part of the Mulago hospital complex, located in Kampala, the capital city of Uganda. Patients with all types of cancers including leukaemia, lymphomas and solid tumours are managed at the UCI; whereas patients with general internal medicine conditions are managed in MNRH. Blood transfusion practices at the two centres generally follow the same guidelines from the Ugandan ministry of health, which recommends blood transfusion based on the clinical condition of the patient, and especially when the haemoglobin level is <7 g/dL or 6 g/dL for patients with sickle cell anaemia. Blood components for the two hospitals are processed and provided by the Uganda Blood Transfusion Services (UBTS), Nakasero-Kampala. Every donated blood products/unit is tested for HIV, hepatitis B, hepatitis C, and syphilis, and can only be issued to the hospital once all the tested serological markers are negative. Nucleic acid test has recently been included at the UBTS to improve transfusion safety. The whole blood units provided for transfusion are preserved in citrate phosphate dextrose adenine (CPDA-1),

kept under refrigerated storage at 1–6°C and are not leucoreduced. At both the UCI and MNRH, pre-transfusion testing consist of recipient's ABO and Rhesus D blood typing, and a room temperature immediate spin cross-match–all performed using the tile method. There is no active haemovigilance system in Uganda.

**Eligibility criteria.** Charts of patients aged >14 years and transfused with whole blood during the months of January, April, July and October of 2018 were purposively sampled. These were quarterly months of the year chosen to account for possible seasonality in supply and transfusion practices. Blood collection is usually at its peak during the seasons when student-donors are in school. Very ill patients in the intensive care units (ICUs) were not included. Charts without demographic identifiers and date of blood transfusion were also not included. The sample size or power calculations were not done a priori.

## Data collection

Data were manually abstracted from the charts using a standardized data abstraction form. Data collected were demographic information, number of blood units received, and mortality status with the date of mortality. Data collected were verified for completeness and accuracy by the principal investigator. Verified data were coded, and entered into a database using Epidata version 3.1 (Epidata association, Denmark) and cleaned to ensure accuracy before exporting to STATA Version 14 (StataCorp, USA) for analysis. Study approvals and waivers of consent were obtained from the Makerere University School of Medicine Research Ethics Committee (Ref. 2017–106) and the Uganda National Council for Science and Technology (Ref. HS 2705). All patients' information was anonymised.

**Data analysis.** Demographic and clinical data were reported using descriptive statistics; using mean and standard deviation for parametric data and median and interquartile range (IQR) for nonparametric data. Patients were retrospectively followed from the time of the first hospital transfusion until death in hospital, or until 30 days when they were administratively censored. Additional blood transfusion episodes were recorded during the follow up period. Patients without evidence of death in the hospital or being alive after 30 days were considered lost to follow up. The 30 day mortality rate was expressed as the ratio of the total number of patients who died within 30 days of transfusion to the total number of patients included in the study. The 30 day overall survival (OS) rate was illustrated using the Kaplan-Meier curve. Survival time was measured in days. Patients who were lost to follow up were included in the analysis and were censored on the last recorded date of review. A Cox proportional hazard model was used to evaluate the association between patient characteristics and mortality. Characteristics with p value <0.05 at univariable analysis and variables that were well known to be associated with mortality, such as the number of blood units transfused were further analysed in a multivariable model. Baseline haemoglobin levels, stages of cancers and HIV characteristics were not included since they were not consistently stated and the staging systems differed among different types of cancers. The global test for the proportional hazard assumption was met (p = 0.93). Hazard ratios and 95% confidence intervals were generated. A two sided statistical significance was set at p<0.05.

## Results

A total of 2,012 transfused patients with a median (IQR) age of 39 (28–54) years were enrolled into the study, of whom 1,107 (55%) were females. The top five common diagnoses were isolated HIV related anaemia, 228 (11.3%); gynaecological cancers, 208 (10.3%); unexplained anaemias, 186 (9.2%); gastrointestinal malignancies, 148 (7.4%); and kidney diseases, 141 (7.0%) (Table 1). Most patients received one unit of whole blood (n = 1,232, 61.2%), and this

**Table 1. Baseline patient characteristics.**

| Diagnosis | Frequency, n | Percentage, % |
|---|---|---|
| Isolated HIV related anaemia | 228 | 11.3 |
| Gynaecological malignancies | 208 | 10.3 |
| Unexplained anaemia | 186 | 9.2 |
| Gastrointestinal cancers | 148 | 7.4 |
| Kidney disease | 141 | 7.0 |
| Leukaemia | 136 | 6.8 |
| Upper gastrointestinal bleeding | 117 | 5.8 |
| Liver disease | 107 | 5.3 |
| Lymphoma | 70 | 3.5 |
| Sickle cell disease | 66 | 3.3 |
| Respiratory disease | 65 | 3.2 |
| Kaposi sarcoma | 63 | 3.1 |
| Cardiovascular disease | 50 | 2.5 |
| Breast cancer | 49 | 2.4 |
| Soft tissue cancer | 46 | 2.3 |
| Diabetes mellitus | 40 | 2.0 |
| Sepsis | 40 | 2.0 |
| Malaria | 39 | 1.9 |
| Urological cancer | 35 | 1.7 |
| Central nervous system disease | 28 | 1.4 |
| Bone marrow failure syndrome | 28 | 1.4 |
| Other gastrointestinal disease | 26 | 1.3 |
| Soft tissue infection | 22 | 1.1 |
| Lung cancer | 13 | 0.7 |
| Venous thromboembolism | 10 | 0.5 |
| Other non-cancers | 9 | 0.5 |
| Bone tumour | 9 | 0.5 |
| Benign gynaecological disease | 8 | 0.4 |
| Bleeding disorders | 6 | 0.3 |
| Trauma | 6 | 0.3 |
| Iron deficiency anaemia | 5 | 0.3 |
| Malnutrition | 4 | 0.2 |
| Other cancers | 4 | 0.2 |

was consistent in all diagnostic categories; followed by 2 units (n = 625, 31.1%) and 3 units (n = 106, 5.3%). Only 49 (2.4%) patients received > 3 units of blood. Data on transfusion with other blood components were not abstracted.

## Mortality rate

Of the 2,012 transfused patients enrolled in the study, 148 died on the same day of transfusion while an additional 359 patients died within 30 days of follow-up. Therefore, the 30 day all-cause mortality rate was 25.2% (507 of 2012 patients). The 30 day overall survival (OS) was 69.3% (95% CI, 66.5–72.4), (**Fig 1**) and the overall median survival was 328 days (47 weeks).

Patients with increased hazards of mortality were those with isolated HIV related anaemia (HR 3.2, 95% CI, 2.3–4.4), liver disease (HR 3.0, 95% CI, 2.0–4.5), kidney disease (HR 2.2, 95% CI, 1.5–3.3; $p<0.01$), cardiovascular diseases (HR 2.9, 95% CI, 1.6–5.4; $p<0.01$), respiratory

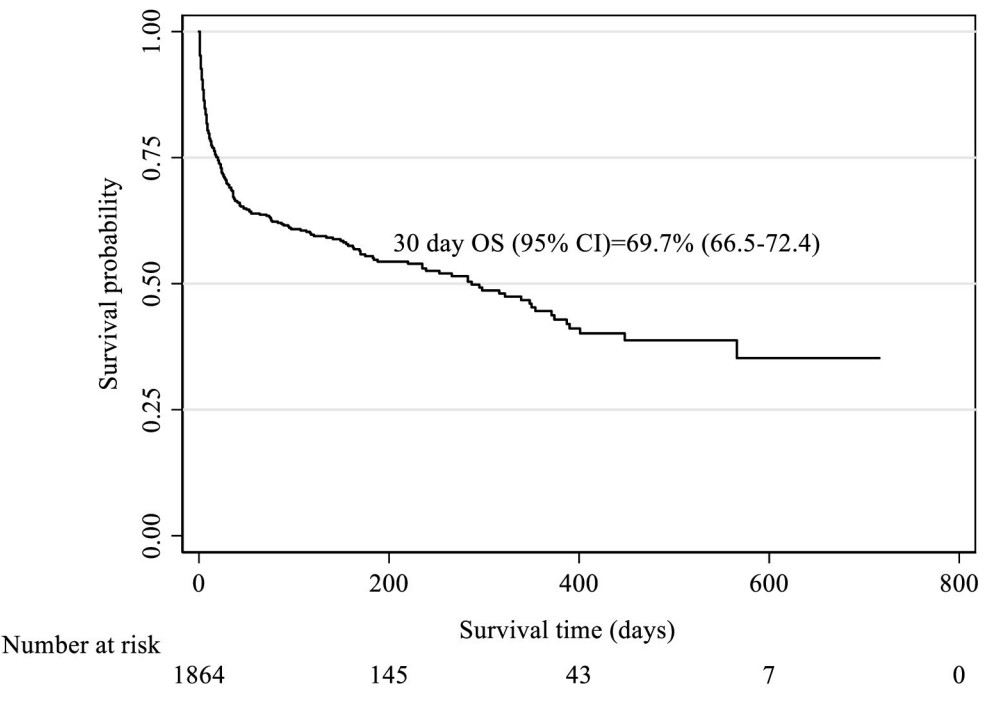

**Fig 1. Kaplan-Meier survival estimate of transfused patients.**

disease (HR 3.0, 95% CI 1.8–4.9; p<0.01), diabetes mellitus (HR 4.1, 95% CI, 2.3–7.4; p<0.01) and sepsis (HR 6.2, 95% CI 3.7–10.4; p<0.01). Patients with reduced hazards of mortality were those who received additional units of blood transfusion (HR 0.8, 95% CI 0.7–0.9; p<0.01) (**Table 2**).

## Discussion

In this retrospective study of patients transfused with whole blood in Uganda, we report a 30 day all-cause mortality rate of 25.2%. Factors associated with mortality were isolated HIV related anaemia, liver disease, kidney disease, cardiovascular disease, respiratory disease, diabetes mellitus and sepsis. Patients in this study were from the general in-patient wards. The mortality rate in this study population was higher than the overall mortality in the general medical patients in Mulago-hospital (17.1%) [21] possibly due to the additional strain of anaemia in the study population.

The majority of patients in our study were transfused with only one unit of blood (61.2%). It was not possible to determine whether the transfusion with only one unit of blood was due to blood supply limitation or physician. If blood supply was an issue, then the mortality rate may suggest inadequate transfusion in patients who might have had severe anaemia. However, haemoglobin thresholds were not abstracted in our study. Anaemia is an independent mortality indicator in older [22] and immunocompromised patients [6]. Moreover, patients who received more blood had reduced mortality in our multivariable analysis (p<0.01).

A causal link between transfusion and mortality is difficult to determine since mortality may be associated with the underlying disease or transfusion itself. A systematic study to assess the association of red blood cell transfusion and in-hospital mortality in patients admitted to the ICU failed to identify association across studies for the number of blood units transfused [23]. Also, a meta-analysis on whole blood transfusion was not associated with in-hospital/

**Table 2. Factors associated with mortality.**

| Characteristic | Univariable analysis | | | Multivariable analysis | | |
|---|---|---|---|---|---|---|
| | HR | p-value | 95% CI | HR | p-value | 95% CI |
| Male sex | 0.8 | 0.02 | 0.7–1.0 | 0.9 | 0.41 | 0.8–1.1 |
| Patient age | 1.0 | 0.03 | 0.9–1.0 | 0.1 | 0.10 | 0.9–1.0 |
| Units of whole blood | 1.0 | 0.58 | 0.8–1.1 | 0.8 | <0.01 | 0.7–0.9 |
| Isolated HIV related anaemia | 2.3 | <0.01 | 1.8–3.1 | 3.2 | <0.01 | 2.3–4.4 |
| Gynaecological malignancies | 0.6 | <0.01 | 0.4–0.8 | 0.7 | 0.07 | 0.5–1.0 |
| Unexplained anaemia | 0.4 | <0.01 | 0.2–0.8 | 0.7 | 0.23 | 0.4–1.3 |
| Gastrointestinal cancers | 0.6 | 0.01 | 0.4–0.9 | 0.8 | 0.16 | 0.5–1.1 |
| Kidney disease | 1.5 | 0.03 | 1.0–2.1 | 2.2 | <0.01 | 1.5–3.3 |
| Leukaemia | 1.2 | 0.37 | 0.9–1.6 | - | - | - |
| Upper gastrointestinal bleeding | 0.8 | 0.55 | 0.5–1.5 | - | - | - |
| Liver disease | 2.1 | <0.01 | 1.4–3.1 | 3.0 | <0.01 | 1.0–4.5 |
| Lymphoma | 0.8 | 0.17 | 0.5–1.1 | - | - | - |
| Sickle cell disease | 0.6 | 0.21 | 0.2–1.4 | - | - | - |
| Respiratory disease | 2.1 | <0.01 | 1.3–3.3 | 3.0 | <0.01 | 1.8–4.9 |
| Kaposi sarcoma | 1.1 | 0.69 | 0.7–1.7 | - | - | - |
| Cardiovascular disease | 1.8 | 0.05 | 1.0–3.3 | 2.9 | <0.01 | 1.6–5.4 |
| Breast cancer | 0.7 | 0.10 | 0.4–1.1 | - | - | - |
| Soft tissue cancer | 0.5 | 0.06 | 0.3–1.0 | - | - | - |
| Diabetes mellitus | 2.7 | <0.01 | 1.5–4.6 | 4.1 | <0.01 | 2.3–7.4 |
| Sepsis | 4.4 | <0.01 | 2.7–7.1 | 6.2 | <0.01 | 3.7–10.4 |
| Malaria | 0.4 | 0.19 | 0.1–1.6 | - | - | - |
| Urological cancer | 0.5 | 0.07 | 0.2–1.1 | - | - | - |
| Central nervous system disease | 1.9 | 0.08 | 0.9–3.8 | - | - | - |
| Bone marrow failure syndrome | 1.4 | 0.35 | 0.7–3.0 | - | - | - |
| Other gastrointestinal disease | 1.4 | 0.47 | 0.6–3.4 | - | - | - |
| Soft tissue infection | 0.2 | 0.08 | 0–1.2 | - | - | - |
| Lung cancer | 0.6 | 0.45 | 0.2–2.0 | - | - | - |
| Venous thromboembolism | 0.6 | 0.62 | 0.1–4.4 | - | - | - |
| Other non-cancers | <0.1 | 1 | 0 -. | - | - | - |
| Bone tumour | 2.3 | 0.06 | 1.0–5.6 | - | - | - |
| Benign gynaecological disease | 1.3 | 0.72 | 0.3–5.2 | - | - | - |
| Bleeding disorders | 1.2 | 0.83 | 0.2–8.9 | - | - | - |
| Trauma | 0.7 | 0.71 | 0.1–4.9 | - | - | - |
| Iron deficiency anaemia | <0.1 | 1 | 0 -. | - | - | - |
| Malnutrition | 1.3 | 0.79 | 0.2–9.4 | - | - | - |
| Other cancers | 0.7 | 0.70 | 0.1–4.9 | - | - | - |

**NOTE:** CI–Confidence Interval; HR–Hazard Ratio.

30-day mortality in patients treated for traumatic haemorrhagic shock [24]. In the setting of massive blood transfusion, a large multicentre retrospective study in China showed lowest mortality when RBCs was transfused at a volume of 5–9 units; however, patient mortality increased with the increase in the volume of $\geq$10 units [25]. A similar finding was also reported in a retrospective study in the USA on 3,523 medical and surgical patients who received blood transfusion, in which mortality increased linearly over the entire dose range with a 10% increase for each 10 units of erythrocytes transfused [26].

Studies that used transfusion threshold to describe 30-day mortality have not even reported consistent results either. Patients transfused with RBCs with a baseline haemoglobin >5.0 mg/dL had more than twice the odds of mortality compared to those with haemoglobin ≤5.0 mg/dL in an emergency setting in Rwanda. However, the small number of patients with haemoglobin level ≤5.0 mg/dL was a limitation in this study [27]. In patients with cancers and septic shock, survival trend favoured those transfused at haemoglobin threshold of <9g/dL [28]. And in a systematic review by Carson et al. [29], transfusion at haemoglobin threshold of <7 – 8g/dL produced similar outcomes with that of haemoglobin 9-10g/dL.

The 30 day mortality in our study is comparable to the Sepsis Occurrence in Acutely Ill Patients (SOAP) study done in Europe, where transfused patients had a mortality rate of 23% [30]. However, most centres (76%) in the SOAP study routinely used leucoreduced RBC without significant supply constraints contrary to our study. Nonetheless, most ICUs have reported lower mortality rates than our study. In a recent meta-analysis that included seven randomized controlled trials (RCTs) from ICUs in Europe [31–33], Canada [34, 35] and Brazil [28, 36], the overall 28 day and 30 day mortality did not exceed 10%. Patients in our study were from the general inpatient wards, but the ICU type patients might have been included due to the scarcity of the ICUs in Uganda [37]. The disease severity score was not reported in our study due to the lack of routinely collected data to calculate these scores.

We observed increased mortality in patients with isolated HIV related anaemia, liver disease, kidney disease, cardiovascular disease, respiratory disease, diabetes mellitus and sepsis. We contend that the association of mortality with the underlying disease may reflect the influence of the disease on mortality [38, 39], rather than transfusion. For example, in a large retrospective study of 50,624 patients in a tertiary hospital in Uganda, the top 5 leading causes of mortality included HIV/AIDS (44.5%), respiratory disease (37.6%), cardiovascular disease (27.8%), sepsis (9%), diabetes mellitus (6.2%) and kidney disease (2.5%) [21], a similar picture to the finding of our study.

The strength of our study is that, it is so far, the first to report the 30-day mortality rate in patients transfused with whole blood in the general medical patient population comprising of cancer and internal medicine patients in the East African region. However, we acknowledge the following weaknesses: firstly, the non-uniform coding of patients' diagnoses and the absence of disease severity score and comorbidities. Secondly, pre-transfusion haemoglobin threshold were not reported and anaemia itself is a strong predictor of mortality. Our study is also unlikely to be generalizable to high income countries that transfuse blood components rather than whole blood.

In conclusion, our result shows the 30 day all-cause mortality in transfused patients to be higher than in the general inpatients, with increased mortality rates in patients with isolated HIV related anaemia, kidney disease, liver disease, respiratory disease, cardiovascular disease, diabetes mellitus and sepsis; however, patients who received additional blood had reduced mortality rates. This finding requires further prospective evaluation.

## Supporting information

**S1 Data.**
(XLSX)

## Acknowledgments

We are indebted to Irene Judith Nassozi and Vivian Bayo for their administrative assistance; Ariko Godfrey Achia, Irene Among, Florence Nantezza and Olivia Nabirye for their invaluable time in data collection.

## Author Contributions

**Conceptualization:** Clement D. Okello, Andrew W. Shih, Nancy Heddle.

**Data curation:** Clement D. Okello, Bridget Angucia.

**Formal analysis:** Bridget Angucia.

**Funding acquisition:** Clement D. Okello.

**Investigation:** Clement D. Okello.

**Methodology:** Clement D. Okello, Andrew W. Shih, Noah Kiwanuka, Nancy Heddle, Harriet Mayanja-Kizza.

**Project administration:** Clement D. Okello.

**Resources:** Jackson Orem.

**Supervision:** Clement D. Okello, Harriet Mayanja-Kizza.

**Validation:** Clement D. Okello.

**Writing – original draft:** Clement D. Okello.

**Writing – review & editing:** Clement D. Okello, Andrew W. Shih, Bridget Angucia, Noah Kiwanuka, Jackson Orem, Harriet Mayanja-Kizza.

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
