## [Decision Letter · Decision Letter 0]

20 Jul 2022

PONE-D-22-16519Mortality and its associated factors in transfused patients at a tertiary hospital in UgandaPLOS ONE

Dear Dr Okello

Thank you for submitting your manuscript to PLOS ONE. After careful consideration, we feel that it has merit but does not fully meet PLOS ONE’s publication criteria as it currently stands. Therefore, we invite you to submit a revised version of the manuscript that addresses the points raised during the review process.

We look forward to receiving your revised manuscript.

Kind regards,

Zivanai Cuthbert Chapanduka, MBChB (M.D)

Academic Editor

PLOS ONE

Journal Requirements:

Additional Editor Comments:

Dear Dr Clement Dove Okello

The reviewers have completed their review of you manuscript and have requested major/minor revisions. Reviewer 1 made the effort to compile all recommendations on a word document. Please attend to each revision to the best of your ability.

Looking forward to receiving your revised manuscript.

Thank you.

Regards

Reviewers' comments:

Reviewer's Responses to Questions

**Comments to the Author**

1. Is the manuscript technically sound, and do the data support the conclusions?

Reviewer #1: Yes

Reviewer #2: Partly

2. Has the statistical analysis been performed appropriately and rigorously? 

Reviewer #1: I Don't Know

Reviewer #2: I Don't Know

3. Have the authors made all data underlying the findings in their manuscript fully available?

Reviewer #1: Yes

Reviewer #2: Yes

4. Is the manuscript presented in an intelligible fashion and written in standard English?

Reviewer #1: Yes

Reviewer #2: Yes

5. Review Comments to the Author

Reviewer #1: Thank you for affording me this opportunity to review this manuscript. I attach hereto a word document containing all my comments in a tabular format. Each comment is aligned with its corresponding author's statement, and referenced using the line numbers.

Reviewer #2: The paper is relevant and adds value on the subject of whole blood transfusions.

Minor Recommendations

Background:

The paper is on whole blood transfusions yet the background is more focused on blood components transfusion. Appreciating there is not much published on whole blood transfusions, the context in which it is commonly used ( eg resuscitation and pediatrics ) could have been included in the introduction. Any papers that have compared whole blood vs blood component transfusions and their conclusions would have added more weight. The comparing of mortality rate of this paper with other published data on patients who were transfused blood components may be challenged.

Study design and setting:

Line 95- the inclusion of surgical wards/trauma would have provided a better comparison with published data

Eligibility criteria:

Line 115- seasonality in supply and transfusion practices. Elaboration on this would be of value as it is a local experience?

Data analysis:

Line 144- the exclusion of baseline hemoglobin, although mentioned as a limitation, is a crucial omission. The degree of anemia is known to be an important risk factor for mortality and a marker of underlying disease.

Mortality rate:

Line 179- the 148 patients who died on the same day of transfusion, further analysis of this group would have been of value and possibly contributed to additional risk factors

6. PLOS authors have the option to publish the peer review history of their article (what does this mean?). If published, this will include your full peer review and any attached files.

Reviewer #1: No

Reviewer #2: No

---

## [Author Response · Author response to Decision Letter 0]

22 Aug 2022

Line 70 “…such as transmission of diseases like hepatitis B and C…” Please use the word “infections” rather than “diseases” ___Addressed and revised whole paragraph

Lines 70-71 “…human 71 immunodeficiency virus.” Please add the abbreviation “HIV” in brackets and use it henceforth.___ Addressed and updated subsequent HIV

Line 71 “…bacteria [2] and malaria [3] as well as…” Please add a comma after malaria.___ Addressed 

Line 77 “…limited blood supply is a major challenge” Is this challenge limited to Sub-Saharan Africa? Or a particular country? Or a continent? Or low-to-middle income countries?____ Specified SSA in line 78

Line 79 “…most blood transfusion in these settings are with blood that is less than 14 days” Is this a good or a bad thing? ___ Addressed in the sentence beginning from line 81 “A randomized, controlled trial conducted in six hospitals in North America, Australia and Israel…”

Line 80 “A randomized, controlled trial conducted in six large hospitals in four countries…” Countries from where? SSA? Africa? Europe? A particular region? Etc. ____ Added North America, Australia and Israel

Lines 81-82 “…increased mortality in patients transfused with fresh red blood cells compared to older red blood cells” What is “fresh” red blood cells? What is “old” red blood cells? ___ Revised to red blood cells stored for a shorter time vs those store for a longer time

Lines 88-89 “We, therefore, undertook a study to describe the mortality and its associated factors in patients transfused with whole blood…” Please confirm that all your study participants received whole blood and not red cell concentrates or other individual blood products._____ Yes we sought out patients who received whole blood and not other blood components

Lines 93-94 “…study conducted at the Uganda Cancer Institute (UCI) and the internal medicine unit of…” It looks like most of your study participants were immunocompromised (HIV and/or malignancy). I would suggest you add a line or so in the background section on the risks of transfusing whole blood in such population.____ Thank you. We have revised paragraph 68-75 to capture this concern

Lines 99-100 “…tumours are treated at the UCI; whereas patients with general internal medicine conditions are taken care of in MNRH I would suggest you use the word “managed” for both “treated” and “taken care of” for uniformity.___ Addressed in lines 101 - 102

Lines 100-101 “…generally follow the same guidelines from the Ugandan ministry of health which recommends…” Please add a comma after the word “health”. ______Addressed, line 103

Line 103 “…haemoglobin level is <7 g/dL or 6 for patients with…” 6 what? Please add the units._____Addressed, line 105

Line 104 “Blood components for the two hospitals are prepared and provided…” Please use the word “processed” instead of “prepared”. _____Addressed, line 106

Lines 105-107 “Prior to delivery to the hospitals, all blood is serologically tested to be negative for human immunodeficiency virus, hepatitis B and C, and syphilis.” Please rephrase: i.e., "Every donated blood products/ unit is tested for HIV, hepatitis B, hepatitis C, and syphilis, and can only be issued to the hospital once all the tested serological markers are negative".

Please ensure that no nucleic acid testing is performed on these units before committing yourself to only serological testing. _____Addressed 

Thank you. Nucleic acid test has recently been included at the UBTS to improve transfusion safety. Lines 109 -110

Lines 110-111 “…immediate spin cross-match – all performed using the tile method which is standard of care in SSA settings” Which one is the standard of care? The time method or just the cross-match? ____Edited out the “standard of care in SSA settings” to avoid confusion

Line 131 “…median and IQR for nonparametric data” Please spell IQR in full the first time. ____Addressed, line 134

Lines 144-145 “Baseline haemoglobin levels, stages of cancers and HIV characteristics were not included since they were not consistently stated…” Baseline haemoglobin could have been a very valuable piece of information in this study. I’m not sure how readily available this information can be from the blood bank? _______We agree this was a big concern. Unfortunately, it is very challenging to again retrieve the patients’ file to obtain this information. Besides, not all files had a baseline haemoglobin as some patients were transfused on clinical grounds. It’s even harder to get them from the hospital blood banks.

Lines 151-152 “The top five common diagnoses were isolated HIV related anaemia…” Parvovirus B19 infection is very common in this group of patients, and requires treatment with IVI immunoglobulins rather than blood transfusion. Is there any information in this regard? _______There was no information on parvovirus B19

Line 153 “…unexplained anaemias, 186 (9.2)…” Please add the units for 9.2 ____Addressed, line 156

Line 154 “Most patients received 1 unit of blood…” Write “1” in words. Write “whole blood” if it is whole blood.__ Addressed, line 157

Line 209 “In this retrospective study of patients transfused with whole blood in Uganda…” Again, please confirm that all your study participants received whole blood and not red cell concentrates or other individual blood products.___ Affirmative 

Line 227 “A systematic study to assess the association of red blood cell transfusion and in-hospital mortality…” You probably might want to also look at the literature about “whole blood” transfusion.____ Sentence added for whole blood on lines 232-234

Line 231 “(red blood cells, RBCs)” Please omit “red blood cells” ____Addressed, line 235

Line 250 “…all patients in our study received non-leucoreduced whole blood.” Please add a line or so in the background about the risks of transfusing “non-leucoreduced whole blood.” ___Addressed in lines 74-75

Lines 260-262 “…the underlying disease condition in our study may simply reflect the inherent influence of the specific disease entity on mortality, rather than the effect of blood transfusion.” I would suggest that the authors add in the background:

- The risks of blood transfusion.

- Advantages and disadvantages of transfusing whole blood vs RBC (leuco-depleted vs unfiltered).

- Clinical impact of anaemia in immunocomromised patients

Please add in the discussion whether your hospitals do have any “haemovigilance” committee, or any designated person who oversees the transfusion-associated reactions. 

First paragraph of background, lines 68-75 has been revised to capture this concern

Thank you we have included information on haemovigilance under study setting, Line 114.

Lines 278-279 “…patients who received additional blood had reduced mortality rates.” Is it possible that “inadequate” transfusion might have been a contributing factor in the higher mortality amongst those participants who received only one unit of whole blood?____ Yes, we have highlighted this in lines 221-222: “…If blood supply was an issue then the mortality rate may suggest inadequate transfusion in patients who might have had severe anaemia…”

---

## [Editor Report · Decision Letter 1]

24 Aug 2022

PONE-D-22-16519R1Mortality and its associated factors in transfused patients at a tertiary hospital in UgandaPLOS ONE

Dear Dr. Okello

Thank you for submitting your manuscript to PLOS ONE. After careful consideration, we feel that it has merit but does not fully meet PLOS ONE’s publication criteria as it currently stands. Therefore, we invite you to submit a revised version of the manuscript that addresses the points raised during the review process.

Please submit your revised manuscript by 7 July 2022 If you will need more time than this to complete your revisions, please reply to this message or contact the journal office at plosone@plos.org. Please include the following items when submitting your revised manuscript:A rebuttal letter that responds to each point raised by the academic editor and reviewer(s). You should upload this letter as a separate file labeled 'Response to Reviewers'.A marked-up copy of your manuscript that highlights changes made to the original version. You should upload this as a separate file labeled 'Revised Manuscript with Track Changes'.An unmarked version of your revised paper without tracked changes. You should upload this as a separate file labeled 'Manuscript'.If applicable, we recommend that you deposit your laboratory protocols in protocols.io to enhance the reproducibility of your results. Protocols.io assigns your protocol its own identifier (DOI) so that it can be cited independently in the future. For instructions see: https://journals.plos.org/plosone/s/submission-guidelines#loc-laboratory-protocols. Additionally, PLOS ONE offers an option for publishing peer-reviewed Lab Protocol articles, which describe protocols hosted on protocols.io. Read more information on sharing protocols at https://plos.org/protocols?utm_medium=editorial-email&utm_source=authorletters&utm_campaign=protocols.

We look forward to receiving your revised manuscript.

Kind regards,

Zivanai Cuthbert Chapanduka, MBChB (M.D)

Academic Editor

PLOS ONE
---

## [Author Response · Author response to Decision Letter 1]

26 Aug 2022

Dear reviewers,

Thank you very much for the speedy response to our submission.

We hereby submit our responses to your comments. Your suggestions are in normal fonts and our responses follow immediately under “Response(s)”.

Journal Requirements:

Response(s): 

• Reference 7, line 74 has been updated. 

• Reference 21, line 108 has been deleted.

Reviewers' comments:

Response(s): No attachments seen.

---

## [Editor Report · Decision Letter 2]

12 Sep 2022

Mortality and its associated factors in transfused patients at a tertiary hospital in Uganda

PONE-D-22-16519R2

Dear Dr. Okello,

We’re pleased to inform you that your manuscript has been judged scientifically suitable for publication and will be formally accepted for publication once it meets all outstanding technical requirements.

Kind regards,

Zivanai Cuthbert Chapanduka, MBChB (M.D)

Academic Editor

PLOS ONE
---

## [Editor Report · Acceptance letter]

13 Sep 2022

PONE-D-22-16519R2 

Mortality and its associated factors in transfused patients at a tertiary hospital in Uganda 

Dear Dr. Okello:

I'm pleased to inform you that your manuscript has been deemed suitable for publication in PLOS ONE. Congratulations! Your manuscript is now with our production department. 

Kind regards, 

on behalf of

Dr. Zivanai Cuthbert Chapanduka 

Academic Editor

PLOS ONE